# COVID-19 Antigen Results Correlate with the Quantity of Replication-Competent SARS-CoV-2 in a Cross-Sectional Study of Ambulatory Adults during the Delta Wave

Yuan-Po Tu,[a] Christopher Green,[b] Linhui Hao,[c] Alexander L. Greninger,[d] Jennifer F. Morton,[e] Heather A. Sights,[a] Michael Gale, Jr.,[c] Paul K. Drain[e]

aThe Everett Clinic – Part of Optum, Everett, Washington, USA

bAbbott Laboratories, Abbott Park, Illinois, USA

cDepartment of Immunology, Center for Innate Immunity and Immune Disease, Center for Emerging & Re-emerging Infectious Diseases, University of Washington, Seattle, Washington, USA

dDepartment of Laboratory Medicine and Pathology, University of Washington, Seattle, Washington, USA

eDepartment of Global Health and International Clinical Research Center, University of Washington, Seattle, Washington, USA

**ABSTRACT** Appropriate interpretation of various diagnostic tests for COVID-19 is critical, yet the association among rapid antigen tests, reverse transcription (RT)-PCR, and viral culture has not been fully defined. To determine whether rapid antigen testing correlates with the presence and quantity of replication-competent severe acute respiratory syndrome coronavirus 2 (SARS-CoV-2) in ambulatory adults, 626 adult participants were enrolled in a cross-sectional diagnostic study. Each participant had two anterior nasal swabs obtained for rapid antigen and RT-PCR testing and SARS-CoV-2 viral culture. The primary outcomes were the presence and quantification of SARS-CoV-2 growth in VeroE6-ACE2-TMPRSS2 cells in asymptomatic and symptomatic ambulatory adults. In this cross-sectional study of 626 adult outpatients, the sensitivity of a single positive antigen test to identify replication-competent SARS-CoV-2 was 63.6% in asymptomatic and 91.0% in symptomatic participants. Viral culture titers were the highest at the onset of symptoms and rapidly declined by 7 days after symptom onset. The positive agreement of the rapid antigen test with RT-PCR at a cycle threshold $C_T$ less than 30 was 66.7% in asymptomatic and 90.7% in symptomatic participants. Among symptomatic participants a with a $C_T$ less than 30, a single antigen test had a positive agreement of 90.7% (95% confidence interval [CI], 84.8% to 94.8%). There was 100% negative agreement as all 425 RT-PCR-negative participants had a negative antigen test. A positive antigen test in symptomatic adults with COVID-19 has a strong correlation with replication-competent SARS-CoV-2. Rapid antigen test results may be a suitable proxy for infectiousness.

**IMPORTANCE** Do rapid antigen test results correlate with replication-competent severe acute respiratory syndrome coronavirus 2 (SARS-CoV-2) (i.e., infectious) virus? In this cross-sectional diagnostic study of 626 adults, the sensitivity of the antigen test to identify replication-competent SARS-CoV-2 was 63.6% in asymptomatic and 91.0% in symptomatic participants. Viral culture titers were the highest at the onset of symptoms and rapidly declined by 7 days after symptom onset. The positive agreement of the rapid antigen test with reverse transcription (RT)-PCR at a $C_T$ of less than 30 was 66.7% in asymptomatic participants and 90.7% in symptomatic participants. A positive antigen test may be an appropriate surrogate for identifying replication-competent virus in symptomatic individuals with COVID-19.

**KEYWORDS** BinaxNOW, COVID-19, COVID-19 antigen, replication-competent, SARS-CoV-2, SARS-CoV-2 viral cultures

Address correspondence to Yuan-Po Tu, ytu@everettclinic.com.

The authors declare a conflict of interest. A.L.G. reports contract testing from Abbott, Cepheid, Novavax, Pfizer, Janssen and Hologic and research support from Gilead and Merck, outside of the described work. C.G. is employed by Abbott Laboratories, Abbott Park, IL. Y-P.T. has received honoraria from Abbott for presentations.

**R**apid antigen tests for severe acute respiratory syndrome coronavirus 2 (SARS-CoV-2) are widely used in the outpatient setting due to their low cost, simplicity, and ease of use (1). Multiple studies have demonstrated the relative sensitivity and specificity of antigen testing compared to reverse transcription (RT)-PCR (2, 3). However, fewer studies have evaluated the performance of rapid antigen testing compared to viral cultures in the outpatient setting (4).

While RT-PCR is a highly sensitive method by which to identify the presence of viral sequences, RT-PCR remains positive long after patients are no longer infectious to others (5). Although studies that demonstrate that viral culture accurately predicts transmissibility of SARS-CoV-2 are lacking, viral culture is a method by which one can identify replication-competent virus, which is necessary for the transmission of infection (6, 7).

Therefore, we sought to define the utility of rapid antigen testing in asymptomatic and symptomatic unvaccinated adults for identifying replication-competent SARS-CoV-2 in viral cultures. The objective was to evaluate the sensitivity of rapid antigen testing compared to viral cultures in asymptomatic and symptomatic adults presenting for SARS-CoV-2 testing in the outpatient setting.

## RESULTS

We screened 637 adult participants, and 11 were considered ineligible: 2 had previously enrolled in the study, 5 did not have a reference RT-PCR result, while 4 had received one or more COVID-19 vaccine doses. Therefore, we enrolled 626 participants into the study, all of whom completed diagnostic testing. Fig. 1 outlines the study and results flow using Standards for Reporting Diagnostic accuracy studies guidelines (8).

The mean age of the participants was 37.5 years (standard deviation [SD], 14.1 years; range, 18 to 84 years), and 320 (51.1%) were male (Table 1). Of the 626 participants enrolled into the study, 210 were asymptomatic, and 416 were symptomatic with a mean of 2.91 days (SD, 1.78 days; range, 1 to 7 days) of symptoms.

**Viral culture results.** SARS-CoV-2 cultures were positive in 166 of 201 (83%) RT-PCR-positive participants, of which 144 were symptomatic and 22 were asymptomatic. Our study was able to recover replication-competent virus in 144 (83.7%) of 172 symptomatic RT-PCR-positive participants and 22 (75.9%) of 29 asymptomatic RT-PCR-positive participants (Table 2). Of 60 randomly selected negative RT-PCR participants, 59 (98%) were viral culture negative (95% confidence interval [CI], 91.1% to 100.0%), which is consistent with a prestudy expectation of 99% probability of culture negative in these participants. The one positive viral culture in the RT-PCR-negative group was asymptomatic and had a tissue culture infectious dose ($TCID_{50}$) of less than 100/mL.

For samples from symptomatic participants that exhibited viral growth, higher $TCID_{50}$ values were strongly correlated with days of symptoms ($P$ value = 0.0006) as shown in Fig. 2. Calculated from the regression slope, the median $TCID_{50}$ was 5,623/mL in symptomatic participants with 1 or 2 days of symptoms and decays by a factor of $2.16\times$/day. The median $TCID_{50}$ is less than 100/mL by 7 days of symptoms, and the regression line intercepts 0 at 9.58 days. Asymptomatic participants had a median $TCID_{50}$ 781/mL.

For the 166 participants that were both culture positive and RT-PCR positive, Fig. 3A compares the antigen-positive ($N = 145$) versus antigen-negative participants ($N = 21$) using a box plot of $TCID_{50}$ values, as well as $C_T$ values (Fig. 3B). The two high outlier participants in the antigen-negative group of the $TCID_{50}$ plot are also the two participants with low $C_T$ values ($C_T$ values of 17.6 and 18.4) in the antigen-negative group of the $C_T$ plot. Antigen-positive participants in this subset had a higher median $TCID_{50}$ of 3,162/mL comparted to the antigen-negative participants with a median $TCID_{50}$ of less than 100/mL. Antigen-positive participants in this subset had a lower median $C_T$ of 17.2 compared to the antigen-negative participants with a median $C_T$ of 27.1 (Fig. 3B).

In the 144 symptomatic RT-PCR-positive participants with a positive viral culture, the $C_T$ value was less than 30 in 141 participants, and only 3 had a $C_T$ value greater than 30. In the 22 asymptomatic RT-PCR-positive participants with a positive viral culture, the $C_T$ value was less than 30 in 20 participants, and only 2 participants had a $C_T$ value greater than 30 (Fig. 1).

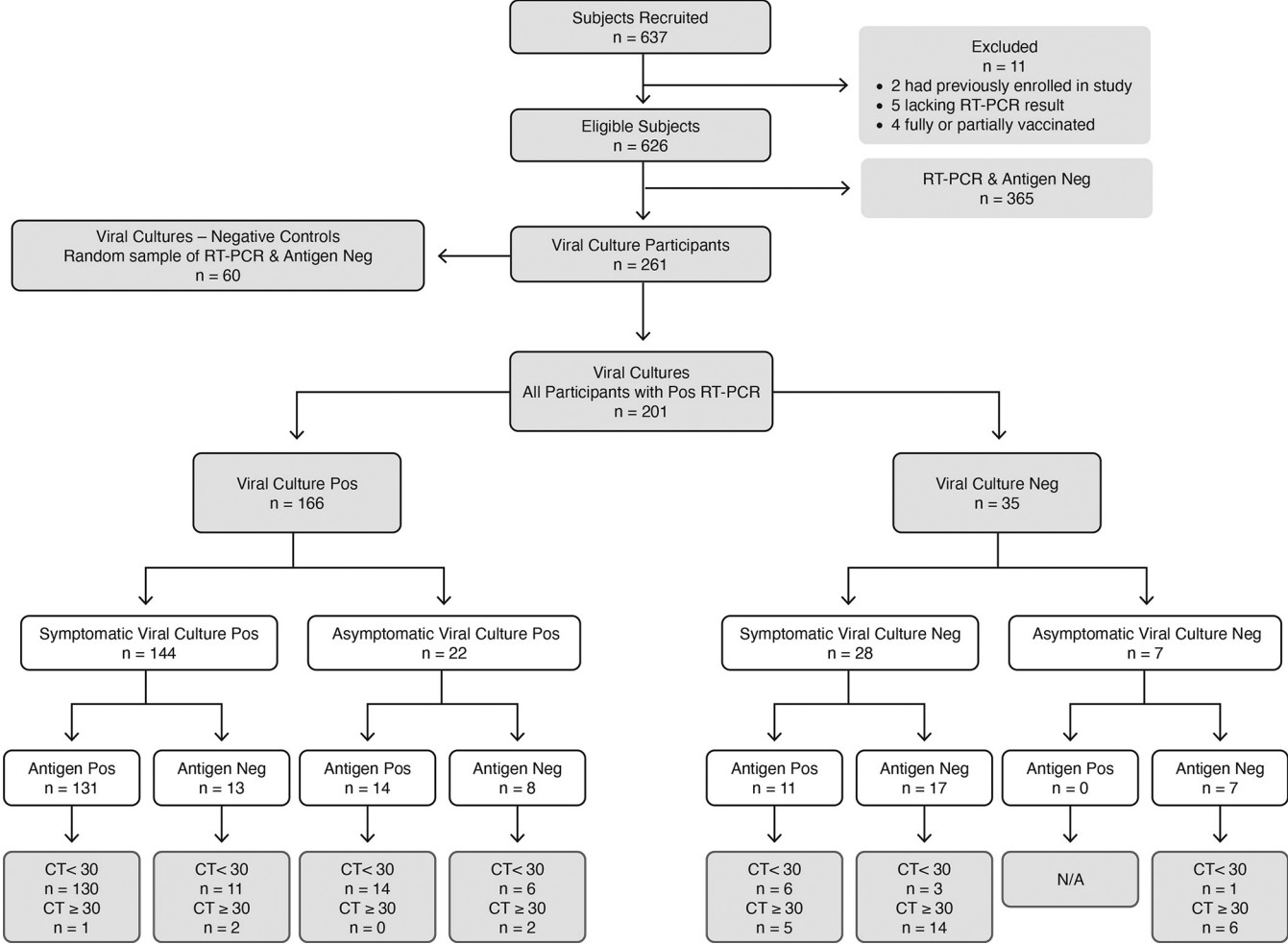

**FIG 1** STAndards for Reporting Diagnostic (STARD) (8) and results flow. STARD accuracy studies diaphragm with reverse transcription (RT)-PCR and antigen results. N/A, not applicable; CT, cycle threshold.

**Antigen and RT-PCR results.** The diagnostic performance of a single antigen test versus RT-PCR is shown in Table 2, in which positive agreement is stratified by the viral culture results. Among asymptomatic participants, antigen testing had a positive agreement of 48.3% (95% CI, 29.4% to 67.5%) compared to RT-PCR, which increased to 63.6% (95% CI, 40.7% to 82.8%) in the subset who were viral culture positive. Among symptomatic participants, a single antigen test had a positive agreement of 82.6% (95% CI, 76.0% to 87.9%) compared to RT-PCR, which increased to 91.0% (95% CI, 85.1% to 95.1%) in the subset of participants who were viral culture positive. Among asymptomatic participants a with a $C_T$ less than 30, antigen testing had a positive agreement of 66.7% (95% CI, 43% to 85.4%), which increased to 90.7% (95% CI, 84.8% to 94.8%) in symptomatic participants. There was 100% negative agreement as all 425 RT-PCR-negative participants had a negative antigen test.

Fig. 4 shows a box plot of $C_T$ values versus symptom status and symptom days. In symptomatic participants, the median $C_T$ indicates RNA copy number peaks on day 3 of symptoms. The positive agreement between a single antigen test and RT-PCR is lower in asymptomatic participants compared to symptomatic participants and increased during the first few days of symptoms, as illustrated in Table 3.

**Logistic regression modeling.** A logistic regression model for predicting a positive viral culture indicates that a lower $C_T$ value has a statistically significant correlation with the probability of viral culture being positive (Fig. 5). If symptom status is included as a second predictor (in addition to $C_T$) of viral culture positivity in the regression model,

**TABLE 1** Characteristics of participants

| Demographic characteristics | Total | Symptomatic | Asymptomatic |
|---|---|---|---|
| No. (%) | 626 (100%) | 416 (66.5%) | 210 (33.5%) |
| **Gender** | | | |
| Female | 306 (48.9%) | 214 (51.4%) | 92 (43.8%) |
| Male | 320 (51.1%) | 202 (48.6%) | 118 (56.2%) |
| **Age (yr)** | | | |
| Mean (SD) | 37.5 (14.1) | 37.4 (14.3) | 37.5 (13.6) |
| Minimum to maximum | 18 to 84 | 18 to 84 | 18 to 76 |
| **Age Categories** | | | |
| 18 to 34 yr (%) | 320 (51.1%) | 211 (50.7%) | 109 (51.9%) |
| 35 to 64 yr (%) | 276 (44.1%) | 184 (44.2%) | 92 (43.8%) |
| 65 to 84 yr (%) | 30 (4.8%) | 21 (5.0%) | 9 (4.3%) |
| **Days symptomatic** | | | |
| $N$ | | 416 | |
| Mean (SD) | | 2.91 (1.78) | |
| Median | | 2 | |
| Minimum to maximum | | 1 to 7 | |

symptom status is a statistically significant predictor ($P$ value = 0.049), and the $C_T$ value at which 50% of viral cultures are positive shifts from 29.0 for symptomatic participants to 33.5 for asymptomatic participants.

The probability of a positive antigen test result as a function of $C_T$ value is shown in Fig. 5 in all 201 PCR-positive participants. If symptom status is included as a second predictor of antigen positivity in the regression model, symptom status is a statistically significant predictor ($P$ value = 0.011), and the $C_T$ value at which 50% of antigen tests are positive shifts from 28.8 for symptomatic participants to 24.0 for asymptomatic participants.

**Viral sequencing results.** Viral sequencing of SARS-CoV-2 was performed using whole-genome sequencing from the viral culture isolates on all 166 positive viral cultures. All the viral isolates were of AY ($n = 164$) and B.1.617.2 ($n = 1$) delta lineage with one exception, which was B.1.1.7 ($n = 1$) from the alpha lineage. The viral sequence of the one positive viral culture in the RT-PCR-negative group was AY from the delta lineage.

**TABLE 2** Diagnostic accuracy of BinaxNOW by viral culture results among symptomatic and asymptomatic adults with SARS-CoV-2 infection[a]

| Culture | RT-PCR | | | RT-PCR positives by $C_T$ category | | RT-PCR positives by viral culture | |
|---|---|---|---|---|---|---|---|
| | Positive | Negative | Total | $C_T < 30$ | $C_T \geq 30$ | Culture positive[b] | Culture negative |
| **Symptomatic participants** | | | | | | | |
| BinaxNOW antigen | | | | | | | |
| Positive | 142 | 0 | 142 | 136 | 6 | 131 | 11 |
| Negative | 30 | 244 | 274 | 14 | 16 | 13 | 17 |
| Total | 172 | 244 | 416 | 150 | 22 | 144 | 28 |
| Positive agreement | | 82.6 (76, 87.9) | | 90.7 (84.8, 94.8) | 27.3 (10.7, 50.2) | 91.0 (85.1, 95.1) | 39.3 (21.5, 59.4) |
| Negative agreement | | 100.0 (98.5, 100.0) | | NA | N/A | NA | N/A |
| **Asymptomatic participants** | | | | | | | |
| BinaxNOW antigen | | | | | | | |
| Positive | 14 | 0 | 14 | 14 | 0 | 14 | 0 |
| Negative | 15 | 181 | 196 | 7 | 8 | 8 | 7 |
| Total | 29 | 181 | 210 | 21 | 8 | 22 | 7 |
| Positive agreement | | 48.3 (29.4, 67.5) | | 66.7 (43, 85.4) | NA | 63.6 (40.7, 82.8) | NA |
| Negative agreement | | 100.0 (98.0, 100.0) | | NA | N/A | NA | N/A |

[a]RT, reverse transcription; SARS-CoV-2, severe acute respiratory syndrome coronavirus 2.
[b]Positive agreement of culture to RT-PCR (considering RT-PCR a reference) is 83.7% (77.3% to 88.9%) for symptomatic participants and 75.9% (56.5% to 89.7%) for asymptomatic participants.

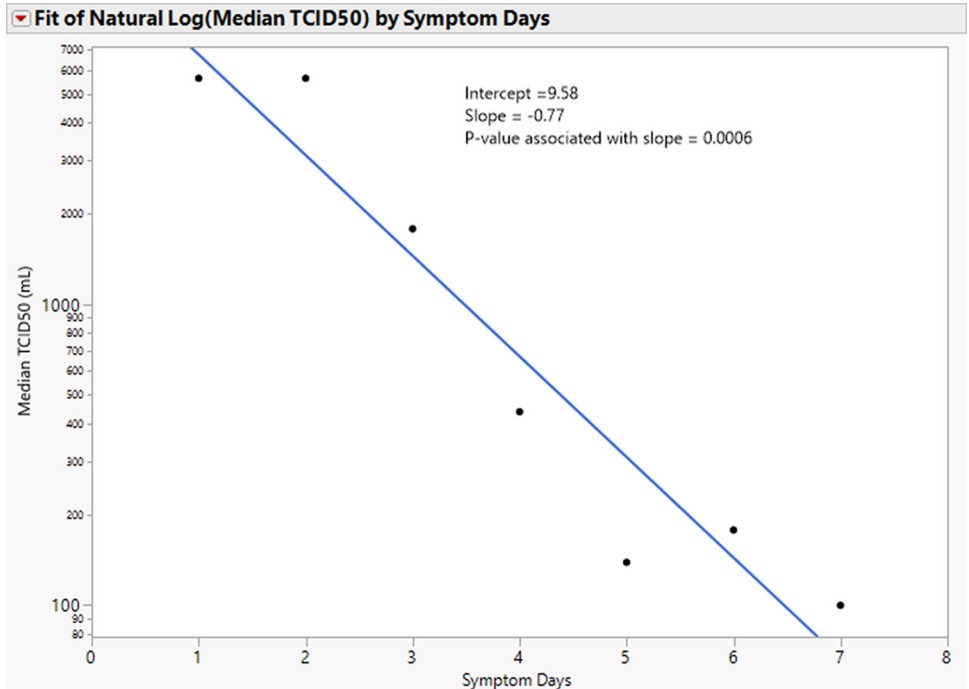

**FIG 2** Tissue culture infectious dose (TCID$_{50}$) versus symptom days. TCID$_{50}$ values are displayed on a log$_{10}$ scale. The blue line is the result of regressing log (median TCID$_{50}$) versus symptom days. The slope is reported in units of natural logarithm per day; i.e., the fold change is 2.16 per day, which is equal to exp(0.77). It is highly statistically significant ($P$ value = 0.0006). Symptomatic participants had a median TCID$_{50}$ less than 100/mL on day 7, and the regression line intercepts 0 at 9.58 days.

## DISCUSSION

In this community-based study, viral cultures demonstrated replication-competent SARS-CoV-2 in RT-PCR-positive participants identified by a single positive rapid antigen test with a sensitivity of 91.0% in symptomatic participants and 63.6% in asymptomatic participants. In this ambulatory cohort, the quantity of viral growth, as measured by TCID$_{50}$, was strongly correlated with higher quantities of viral RNA (low $C_T$), as measured by RT-PCR. Of participants with replication-competent virus, 97% had $C_T$ values of <30 cycles. Overall,

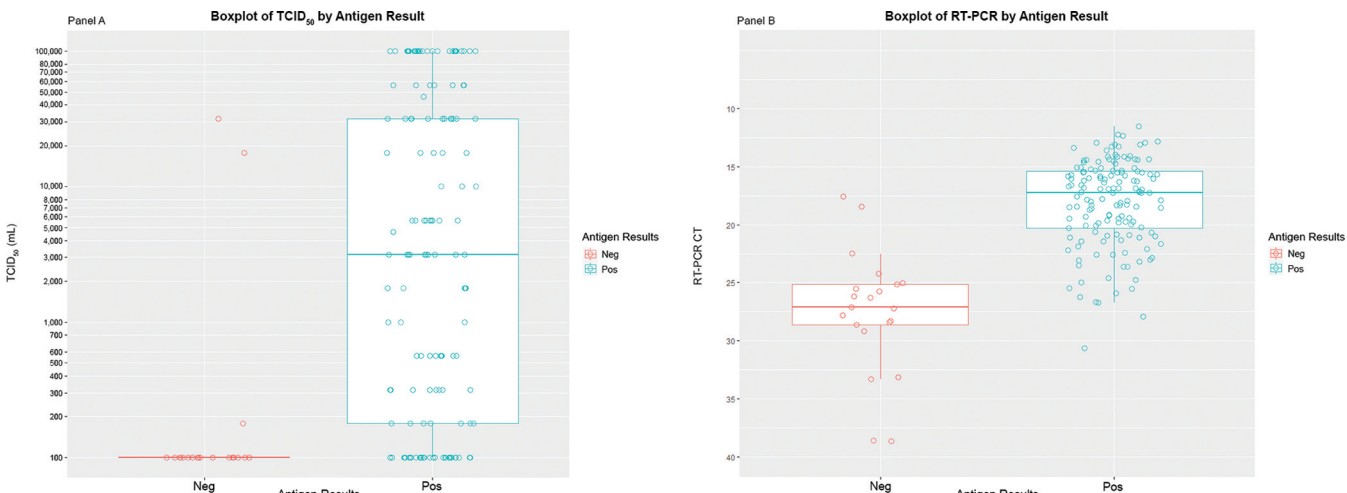

**FIG 3** TCID$_{50}$ and RT-PCR $C_T$ results versus antigen. Comparison of antigen-positive versus antigen-negative participants among the subset of participants that are both culture positive and RT-PCR positive. Antigen-positive and antigen-negative participants are compared using a box plot of TCID$_{50}$ values (A), as well as $C_T$ values (B). The box plot of the antigen-negative group in panel A is collapsed to a line, as there are 18 participants who had a TCID$_{50}$ of 100. There were 21 participants in the antigen-negative group and 145 participants in the antigen-positive group. The two high outlier participants in the antigen-negative group of the TCID$_{50}$ plot are also the two participants with low $C_T$ ($C_T$ values of 17.6 and 18.4) values in the antigen-negative group of the $C_T$ plot.

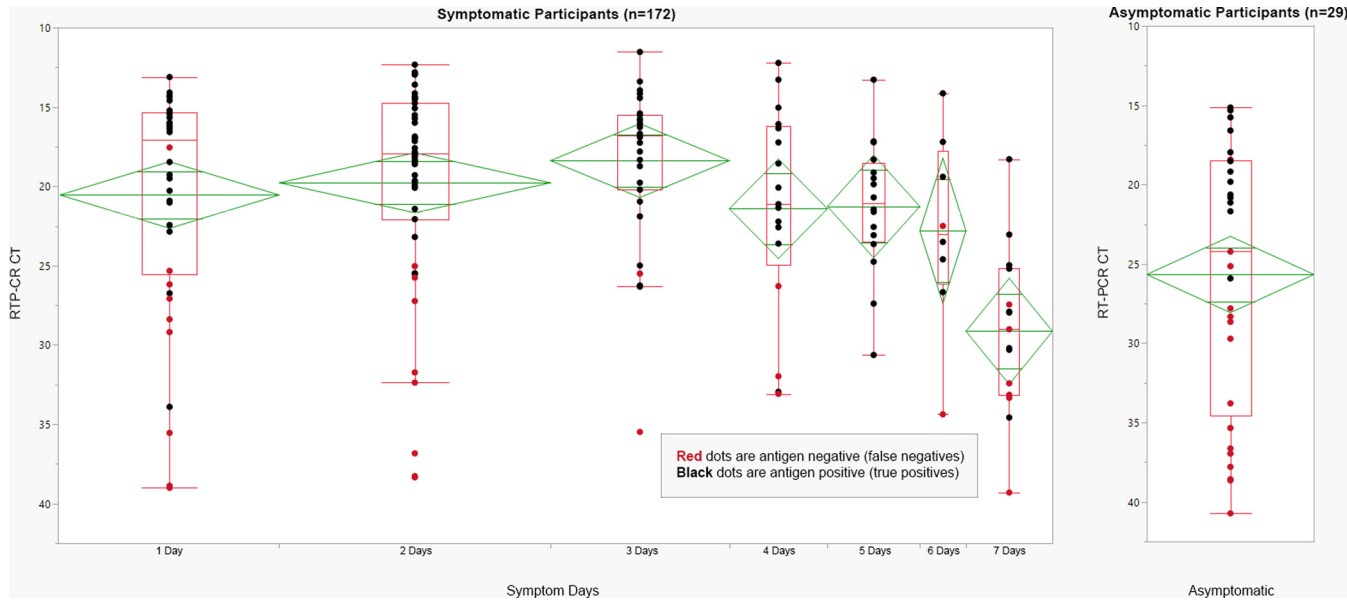

**FIG 4** $C_T$ and antigen results versus symptom days. Box plots of RT-PCR $C_T$ values sorted by days of symptoms. Symptomatic participants ($n = 172$) are sorted by days of symptoms. Asymptomatic participants ($n = 29$) are displayed on the right side of the figure. The red dots are false negatives, for which the antigen test was negative and the RT-PCR was positive. The black dots are true positives, for which both the antigen test and the RT-PCR were positive. $C_T$ values are plotted from the lowest RNA copy number to the highest RNA copy number on the vertical axis.

these results suggest that rapid antigen testing in symptomatic adults with COVID-19 is generally correlated with the presence of replication-competent SARS-CoV-2 and may serve as a potential proxy measure for infectiousness.

This study adds to previous studies demonstrating a high degree of correlation between a positive antigen test and a positive viral culture with lower $C_T$ values (4, 9, 10). While many studies (4) have analyzed antigen test positive agreement to RT-PCR based $C_T$ value, this study provides additional viral culture evidence for this stratification. In symptomatic participants, the $C_T$ value at 50% probability of viral culture positive is nearly the same as the $C_T$ value at 50% probability of antigen positive: 29.0 and 28.8, respectively. This outcome supports using a $C_T$ value of greater than 30 as having a low probability of being culture positive.

Overall, our study enrolled fewer asymptomatic RT-PCR-positive participants. In asymptomatic participants with a $C_T$ value between 24 and 30, the antigen test is more likely to be negative compared to viral culture. While asymptomatic participants with a high $C_T$ value may have little clinical significance, some asymptomatic participants had a low $C_T$ value and

**TABLE 3** Percentage of positive agreement for antigen versus RT-PCR by symptom days and median $C_T$ values[a]

| Days symptomatic | False negatives | True positives | Total | Positive agreement (%) | Median $C_T$ value and IQ range | | |
|---|---|---|---|---|---|---|---|
| | | | | | 25% | Median | 75% |
| 1 day | 9 | 29 | 38 | 76.3 | 15.4 | 17.1 | 25.5 |
| 2 days | 8 | 39 | 47 | 83.0 | 14.8 | 17.9 | 22.1 |
| 3 days | 2 | 29 | 31 | 93.5 | 15.5 | 16.8 | 20.2 |
| 4 days | 3 | 14 | 17 | 82.4 | 16.2 | 21.1 | 25.0 |
| 5 days | 0 | 16 | 16 | 100.0 | 18.5 | 21.1 | 23.5 |
| 6 days | 2 | 6 | 8 | 75.0 | 17.8 | 23.0 | 26.2 |
| 7 days | 6 | 9 | 15 | 60.0 | 25.2 | 29.0 | 33.1 |
| Symptomatic totals | 30 | 142 | 172 | | | | |
| Asymptomatic | 15 | 14 | 29 | 48.3 | 18.5 | 24.2 | 34.6 |

[a]The table shows positive agreement between a single antigen test and RT-PCR results and the median $C_T$ value and interquartile range of symptomatic participants by days of symptoms and asymptomatic participants.

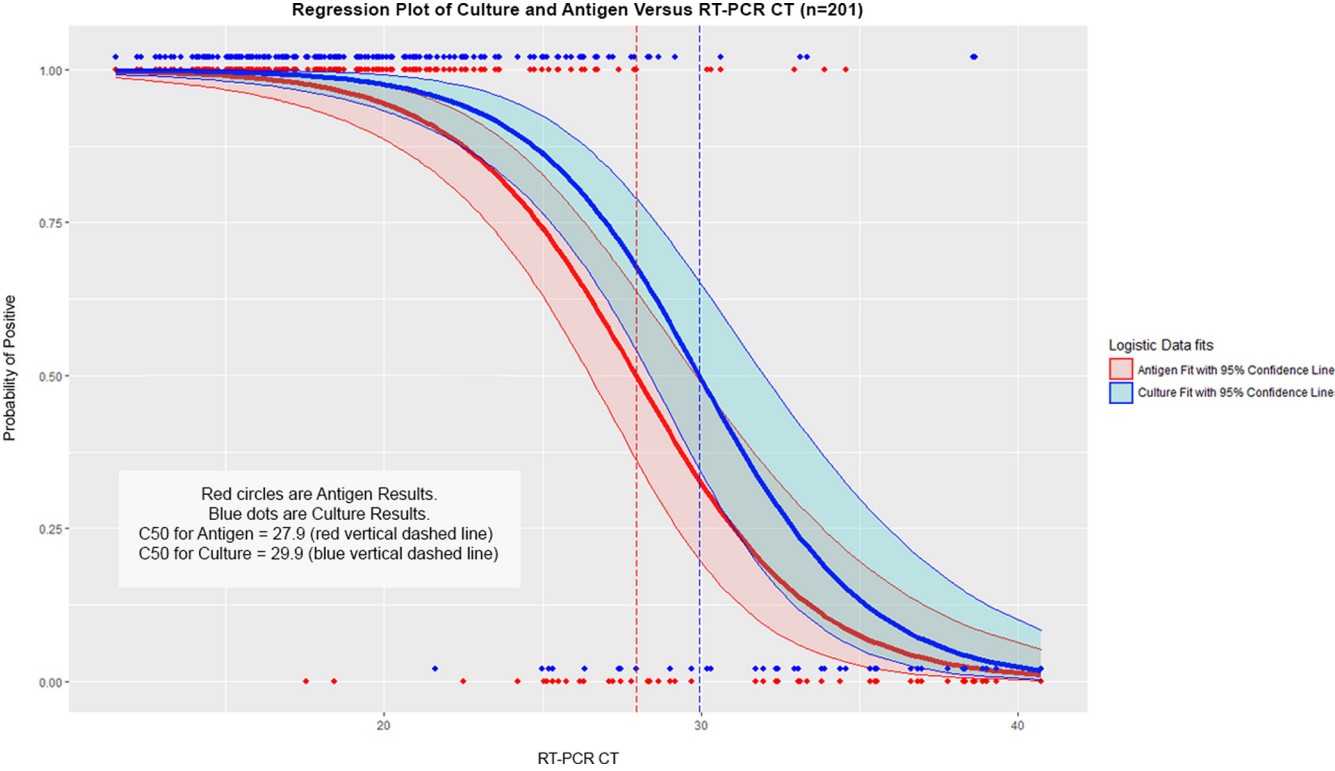

**FIG 5** Probability of culture positive versus $C_T$ value and antigen positive versus $C_T$ value. The red line corresponds to the fit of antigen versus $C_T$, and the blue line corresponds to the fit of culture versus $C_T$. The shaded areas are the 95% confidence limits of the regression. The C50 (represented by the vertical dashed lines) is the $C_T$ value for which there is a 50% chance of being positive based on a given regression line. The $C_T$ value has a statistically significant correlation with the probability of both culture positivity and antigen positivity within the context of the logistic model. A lower $C_T$ value predicts a higher probability of antigen positivity and a higher probability of culture positivity.

negative antigen test but were viral culture positive. Based on clinical follow-up, some of these asymptomatic participants had presented prior to the onset of symptoms. However, due to the small sample size of the asymptomatic RT-PCR-positive subset and incomplete clinical follow-up, the study was unable to stratify the presymptomatic participants. These data support the FDA recommendation (11) and CDC guidance (12) for repeat antigen testing at least twice over 3 days in symptomatic individuals with a negative home antigen test and serial testing at least three times over 5 days for individuals without symptoms of COVID-19, to reduce the risk of a false-negative antigen result.

The quantity of viral culture growth as measured by $TCID_{50}$ was highest in participants during the first 2 days of symptoms and rapidly declined to a $TCID_{50}$ less than 100/mL by day 7 of symptoms. These results indicate that infected individuals have a much higher load of replication-competent virus and are likely to be the most contagious early in the course of their disease.

**Strengths and limitations.** A major strength of this study was the large number of SARS-CoV-2 viral cultures performed, including all 201 RT-PCR-positive participants and a random selection of 60 RT-PCR-negative controls. Our study was novel in that in addition to viral cultures, we measured the relative quantity of replication-competent virus as expressed by $TCID_{50}$. Our study participants were outpatients with mild disease, which represents the majority of community-based SARS-CoV-2 testing and did not rely on convenience sampling methods (13); nor did we include severely ill or hospitalized participants (14, 15).

This study has limitations. The study enrolled participants at a single center who were unvaccinated and nearly all the infections were of AY delta lineage. The study assessed the performance of only a single rapid antigen test using BinaxNOW and did not perform sequential testing on participants with a negative antigen test result.

Although replication-competent virus is necessary for transmission, the lack of the ability to isolate virus does not necessarily imply lack of infectiousness (16). The degree of infectivity

of persons with low viral loads as expressed by low TCID$_{50}$ values remains an important area of future research.

**Conclusions.** A single positive rapid antigen test identifies replication-competent SARS-CoV-2 virus in the vast majority (91.0%) (95% CI 85.1%, 95.1%) of symptomatic adults and most asymptomatic adults in this study conducted during the delta wave. Viral culture titers were the highest at the onset of symptoms and rapidly declined over the following 7 days. A positive rapid antigen test may therefore be an appropriate surrogate for identifying replication-competent virus in symptomatic individuals with COVID-19.

**Institutional review board approval.** All procedures performed in this study involving human participants were in accordance with the ethical standards of the institutional and/or national research committees and with the 1964 Helsinki declaration and its later amendments or comparable ethical standards. Prior to the start of data collection, the study received ethical approval from the Institutional Review Board by the UnitedHealth Group, Office of Human Research Affairs 2021-0032, federal assurance No. FWA00028881, and Office for Human Research Protections (OHRP) Registration No. IORG0010356.

## MATERIALS AND METHODS

**Study design and oversight.** A diagnostic validation study was conducted from June 1, 2021, through September 16, 2021, in Snohomish County, Washington, where the first case of COVID-19 was diagnosed in the United States (17). Potential participants were recruited as they presented to a COVID-19 drive-through testing site and three Everett Clinic urgent care facilities.

Adults were eligible for inclusion if they were unvaccinated adults 18 years of age or older seeking SARS-CoV-2 testing who were willing and able to provide informed consent. Eligible participants could be either asymptomatic or symptomatic with less than 8 days of symptoms. We defined symptomatic cases as individuals who reported having a COVID-19-related symptom, including cough, sore throat, congestion, fever, loss of taste or smell, nausea, vomiting, or diarrhea. Asymptomatic individuals denied having any of the above symptoms.

Participants were excluded if they were hospitalized; had 8 days or more of COVID-19 symptoms; had tested positive for SARS-CoV-2 in the previous 90 days; previously participated in the study; requested testing for travel clearance by a commercial airline or destination or preprocedural testing; had an active nosebleed; had recent facial injuries/trauma or nasal surgery; had previously received a COVID-19 vaccine or were enrolled in a study to evaluate any investigational drug or vaccine; or were unable or unwilling to provide informed consent. Participants who met the inclusion requirements and did not have exclusion criteria were included in the data analysis. After informed consent, basic information was collected, including age, sex, COVID-19 vaccination history, and the number of days of symptoms using a standardized form approved for use by the UnitedHealth Group Institutional Review Board.

**Specimen collection and testing.** Participants had two anterior foam nasal swabs (Puritan Foam Swab 25-1506 1PF 100, Guilford, ME, USA) collected by trained investigators using a crossover technique to maximize the uniformity as previously described (18). One swab was tested immediately on site with the Abbott BinaxNOW COVID-19 antigen (Ag) card (Abbott, Scarborough, ME, USA), according to the manufacturer's instructions for use. External negative and positive controls were performed daily. Photographs of the BinaxNOW COVID-19 Ag card were obtained to validate the fidelity of the data used for analysis. Participants were informed of their antigen test results prior to the conclusion of their study visit. The second swab was eluted into 3 mL of viral transport medium (VTM) (Remel R12701, ThermoFisher Scientific, Hampshire, UK), stored, and transported at 4°C prior to RT-PCR testing.

Nasal swabs were tested for SARS-CoV-2 RNA by RT-PCR using the Abbott Alinity m SARS-CoV-2 assay, a dual target assay for the RdRp and N genes, following the instructions for use (https://www.fda.gov/media/137979/download). The clinical performance of the Abbott Alinity m SARS-CoV-2 assay has been previously evaluated and compared with the Roche Cobas 6800 and Panther Fusion platforms (19). If the initial test sample produced an invalid RT-PCR result, a second VTM aliquot was used to repeat the RT-PCR testing. Following RT-PCR testing, 500-$\mu$L residual VTM aliquots were stored at $-65°$ or colder prior to viral cultures. Participants with a negative antigen test and positive RT-PCR result were informed of their positive results.

**SARS-CoV-2 cultures.** All participants with a positive RT-PCR ($n = 201$) and a subset of 60 randomly selected participants from the 425 participants with a negative RT-PCR and negative antigen results had viral cultures performed on the VTM samples in a Biosafety Lab-3 (BSL-3) facility. All 261 viral cultures were performed at the same time to minimize batch variation.

The VTM samples were filtered through Corning Costar Spin-X centrifuge tube filter (CLS8160), and 0.1 mL was used to infect VeroE6AT cells ectopically expressing human angiotensin-converting enzyme 2 (ACE2) and human transmembrane serine protease 2 (TMPRSS2) (gifted from Barney Graham, National Institutes of Health, Bethesda, MD) in triplicate. The infected cells were examined daily for syncytium formation followed by cell death for up to 10 days. Culture medium was collected when 90% of the cells in the well were dead. The titers of samples with positive virus growth were further determined for a median tissue culture infectious dose (TCID$_{50}$) value. The assay was performed with VeroE6AT cells in 96-well plates. Then 10 $\mu$L of filtered VTM was added with 90 $\mu$L of a postinfection medium as the starting point; 10-fold serial dilutions were performed from the second through the fourth wells. Each sample was tested in four replicates. At 3 days postinfection, the cells were fixed with 10% formaldehyde and stained with 1% crystal violet.

Wells with dead cells were scored to calculate $TCID_{50}$. Samples with negative virus growth were assayed the same way without further dilution points to confirm the growth phenotype. The viral culture methods were previously described in detail (20).

**Whole-genome sequencing of viral RNA.** An aliquot of the collected virus culture from the growth assay was subject to RNA extraction (Zymo Research, R1040) and sequencing using the Swift SARS-CoV148 2 SNAP version 2.0 kit (Swift Biosciences, Ann Arbor, MI, USA) on Illumina NextSeq 500 (Illumina, San Diego, CA, USA). Consensus genome sequence were generated through the covid-swift-pipeline (https://github.com/greninger-lab/covid_swift_pipeline), and the lineages were assigned based on the Pangolin dynamic lineage nomenclature scheme (21).

**Data analysis.** We calculated the positive and negative agreement of the rapid antigen test versus RT-PCR in asymptomatic and symptomatic participants. We stratified the positive agreement of the antigen test into two subgroups by RT-PCR $C_T$ value ($C_T < 30$ and $C_T \geq 30$). We also stratified the positive agreement of the antigen test into two subgroups by SARS-CoV-2 viral culture (positive and negative). Logistic regression was performed to estimate the probability of an antigen-positive test result versus $C_T$ value, as well as the probability of a viral culture positive test results versus the $C_T$ value.

We also examined RT-PCR $C_T$ values and evaluated the $TCID_{50}$ in the viral culture samples. We characterized the distribution (mean, median, SD, interquartile range) of $C_T$ values and the distribution of $TCID_{50}$ viral titers by days of symptoms. We calculated the rate of rate of decline of $TCID_{50}$ viral titers by regressing the log $TCID_{50}$ viral titers for each day of symptoms.

Antigen-positive versus antigen-negative participants were compared among the subset of participants that were both culture positive and PCR positive using a box plot of $TCID_{50}$ and $C_T$ values. We reported the viral lineage from all the positive viral cultures. We used JMP 14.0.0 and R 4.1.1 software for statistical analysis.

**Data availability.** The full genome viral sequences have been posted to the NCBI GenBank (GISAID identifier EPI_SET_230315xa, doi: 10.55876/gis8.230315xa). A data set including results of antigen, RT-PCR $C_T$, viral culture results, viral lineage, and symptom status with days of symptoms will be provided upon reasonable request and a data sharing agreement by contacting the corresponding author.

## ACKNOWLEDGMENTS

We acknowledge the contributions of Ken Kupfer, Amy S. Broulik, and Philip J Ginsburg from Abbott; Lorraine Bell, Susan Spanos, Anne Hartman, Amanda Wells, Kim Gangloff, medical assistants, staff, and providers from the Everett Clinic; and especially the patients who graciously contributed to this study.

A.L.G. reports contract testing from Abbott, Cepheid, Novavax, Pfizer, Janssen, and Hologic and research support from Gilead and Merck, outside the described work. C.G. is employed by Abbott Laboratories (Abbott Park, IL). Y-P.T. has received honoraria from Abbott for presentations.

Abbott Diagnostic Scarborough, Inc. (Abbott), provided funding and the BinaxNOW COVID-19 Ag card and Alinity PCR test kits for the following investigator-initiated study: A Study Comparing Antigen Testing to Standard RT-PCR and Viral Cultures for the Detection of SARS-CoV-2 and Collecting Blood Samples for Research under Protocol No. 2031001. M.G. was supported by National Institutes of Health, NIAID grant AI151698. P.K.D. and J.F.M. received grant INV-017205 from the Bill and Melinda Gates Foundation.

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
