## [Reviewer comments · Microbiology Spectrum]

Microbiology Spectrum

COVID-19 antigen results correlate with the quantity of replication-competent SARS-CoV-2 in a cross-sectional study of ambulatory adults during the delta wave.

Yuan-Po Tu, Christopher Green, Linhui Hao, Alexander L. Greninger, Jennifer Morton, Heather Sights, Michael Gale, Jr., and Paul Drain

Corresponding Author(s): Yuan-Po Tu, The Everett Clinic

Review Timeline:

Submission Date:	January 5, 2023
Editorial Decision:	March 6, 2023
Revision Received:	March 17, 2023
Accepted:	March 28, 2023

Editor: Oliver Laeyendecker

Reviewer(s): Disclosure of reviewer identity is with reference to reviewer comments included in decision letter(s). The following individuals involved in review of your submission have agreed to reveal their identity: Denis E. Kainov (Reviewer #1); Daniel Griffin (Reviewer #2)

Transaction Report:

DOI: <https://doi.org/10.1128/spectrum.00064-23>

March 6, 2023

Dr. Yuan-Po Tu
The Everett Clinic
Urgent Care
3901 Hoyt
Everett, WA 98201

Re: Spectrum00064-23 (COVID-19 antigen results correlate with the quantity of replication-competent SARS-CoV-2 in a cross-sectional study of ambulatory adults)

Dear Dr. Yuan-Po Tu:

Link Not Available

Sincerely,

Oliver Laeyendecker

Journals Department
Reviewer comments:

Reviewer #1 (Comments for the Author):

1. Of the 166 viral isolates in the study, 164 belonged to the AY lineage. Thus, the authors can only state that the results obtained with their rapid antigen test correlate with replication-competent SARS-CoV-2 strains of AY lineage. So the title and conclusions should be more specific.
2. Please deposit full genome viral sequences in public repository and provide accession numbers.
3. Please provide primers used in RT-PCR.
4. Please sequence several full genomes prior virus propagation in cell culture. Comparison of the genomes prior and after virus propagation would allow you to estimate the effect of virus culture.

Reviewer #2 (Comments for the Author):

This is an excellent and well written paper with which I see no significant issues. I think the paper as is is fine. This paper should add to the literature and is strengthened by moving past the simple binary and including TCID₅₀ as well as CT values. With this said the only modification is regarding the language where RNA copies are referred to as viral load. I would suggest that PCR which measures RNA copy number be properly referred to as such as this paper demonstrates a correlation between RNA copy number but also by doing quantitative viral culture acknowledges that PCR measures RNA copy number and not viral load.

Staff Comments:

Preparing Revision Guidelines

Please return the manuscript within 60 days; if you cannot complete the modification within this time period, please contact me. If you do not wish to modify the manuscript and prefer to submit it to another journal, please notify me of your decision immediately so that the manuscript may be formally withdrawn from consideration by Microbiology Spectrum.

Response to Reviewers

Reviewer #1 (Comments for the Author):

1. Of the 166 viral isolates in the study, 164 belonged to the AY lineage. Thus, the authors can only state that the results obtained with their rapid antigen test correlate with replication-competent SARS-CoV-2 strains of AY lineage. So the title and conclusions should be more specific.
2. Please deposit full genome viral sequences in public repository and provide accession numbers.
3. Please provide primers used in RT-PCR.
4. Please sequence several full genomes prior virus propagation in cell culture. Comparison of the genomes prior and after virus propagation would allow you to estimate the effect of virus culture.

Response to Reviewer #1:

We thank the reviewer for his/her dedication to a detailed review, and we greatly appreciated the constructive suggestions given that have helped strengthen the overall manuscript.

1. Of the 166 viral isolates in the study, 164 belonged to the AY lineage. Thus, the authors can only state that the results obtained with their rapid antigen test correlate with replication-competent SARS-CoV-2 strains of AY lineage. So the title and conclusions should be more specific.

Response: To more accurately reflect that the study was performed during the delta wave, we have changed the title to:

COVID-19 antigen results correlate with the quantity of replication-competent SARS-CoV-2 in a cross-sectional study of ambulatory adults during the delta wave.

In the manuscript we have clarified that the viral sequences show nearly all the positive subjects were infected with delta variant AY lineage. (Revisions are highlighted in yellow). The relevant section of the discussion under strengths and limitation and conclusion now reads:

Strengths and Limitations

A major strength of this study was the large number of SARS-CoV-2 viral cultures performed, including all 201 RT-PCR positive participants and a random selection of 60 RT-PCR negative controls. Our study was novel in that in addition to viral cultures, we measured the relative quantity of replication-competent virus as expressed by TCID₅₀. Our study participants were outpatients with mild disease, which represents the majority of community-based SARS-CoV-2 testing and did not rely on convenience sampling methods¹⁸ nor did we include severely ill or hospitalized participants^{19,20}.

This study has limitations. The study enrolled participants at a single center who were unvaccinated and nearly all the infections were of **AY delta lineage**. The study only assessed the performance of a single rapid antigen test using BinaxNOW and did not perform sequential testing on participants with a negative antigen test result.

Although replication-competent virus is necessary for transmission, lack of the ability to isolate virus does not necessarily imply lack of infectiousness²¹. The degree of infectivity of persons with low viral loads as expressed by low TCID₅₀ values remains an important area of future research.

Conclusions

A single positive rapid antigen test identifies replication-competent SARS-CoV-2 virus in the vast majority (91.0%) (95% CI 85.1%, 95.1%) of symptomatic adults and most asymptomatic adults **in this study conducted during the delta wave**. Viral culture titers were the highest at the onset of symptoms and rapidly declined over the following seven days. A positive rapid antigen test may therefore be an appropriate surrogate for identifying replication-competent virus in symptomatic individuals with COVID-19.

2. Please deposit full genome viral sequences in public repository and provide accession numbers.

Response: The full genome viral sequences have been posted to the NCBI GenBak (GISAID). GISAID Identifier: EPI_SET_230315xa. doi: [10.55876/gis8.230315xa](https://doi.org/10.55876/gis8.230315xa).

The manuscript has been revised in viral sequencing section. The revision highlighted in yellow now reads:

Viral sequencing results

Viral sequencing of SARS-CoV-2 was performed using whole genome sequencing from the viral culture isolates on all 166 positive viral cultures. All the viral isolates were of AY (n=164) and B.1.617.2 (n=1) delta lineage with one exception which was B.1.1.7 (n=1) from the alpha lineage. **The full genome viral sequences have been posted to the NCBI GenBak (GISAID). GISAID Identifier: EPI_SET_230315xa. doi: [10.55876/gis8.230315xa](https://doi.org/10.55876/gis8.230315xa)**

3. Please provide primers used in RT-PCR.

Response: The Abbott Alinity m SARS-CoV-2 was used to perform the RT-PCR assays in this study. There are no references to the exact RT-PCR primer sequence in the public domain as the manufacture consider the primer sequence proprietary. We have asked the manufacture for permission to include additional information regarding the primer sequence. The manufacture has referenced the Instructions for Use (IFU) (<https://www.fda.gov/media/137979/download>). The IFU states "The Alinity m SARS-CoV-2 assay is a dual target assay for the RdRp and N genes".

The Alinity M SARS-CoV-2 assays were performed at the University of Washington in the laboratory of Dr. Alex Greninger, who is a co-author on this study. The manuscript references a study conducted in the Dr Alex Greninger laboratory (reference # 10 in the manuscript: PMID 34023572. <https://pubmed.ncbi.nlm.nih.gov/34023572/>) which compares the performance characteristic of various commercially available SARS-CoV-2 assay.

The manuscript has been revised in the Specimen collection and testing section. The revision highlighted in yellow now reads:

Nasal swabs were tested for SARS-CoV-2 ribonucleic acid (RNA) by RT-PCR using the Abbott Alinity m SARS-CoV-2 Assay, **a dual target assay for the RdRp and N genes**, following the **instructions for use** (<https://www.fda.gov/media/137979/download>).

4. Please sequence several full genomes prior virus propagation in cell culture. Comparison of the genomes prior and after virus propagation would allow you to estimate the effect of virus culture.

Response: We greatly appreciate reviewer #1 comments that having the full viral genome from the original viral transport media (VTM) sample and comparing it to the full viral genome of the virus after propagation in viral cultures would be insightful. Sequencing the primary VTM sample was outside the scope and funding of our study plan. We have stored the original residual VTM samples and would like to conduct additional studies possibly including sequencing of the original VTM samples, viral neutralization, and host immune response. We are hopeful that we will be successful in obtaining the necessary funding to perform these important studies.

While we cannot completely rule out the possibility of new laboratory-adapted quasi-species being generated in the P1 virus cultures from the original patient isolate VTM, our group has been isolating and sequencing virus from SARS-CoV2 infected individuals since the inception of virus surveillance efforts in Seattle. Through these efforts, we have completed numerous comparisons of virus sequenced from VTM compared to the P1 stock generated from the VTM, and so far, we have not seen any significant changes in sequence between virus in the VTM and the P1 culture.

Reviewer #2 (Comments for the Author):

This is an excellent and well written paper with which I see no significant issues. I think the paper as is fine. This paper should add to the literature and is strengthened by moving past the simple binary and including TCID50 as well as CT values. With this said the only modification is regarding the language where RNA copies are referred to as viral load. I would suggest that PCR which measures RNA copy number be proper referred to as such as this paper demonstrates a correlation between RNA copy number but also by doing quantitative viral culture acknowledges that PCR measures RNA copy number and not viral load.

Response to Reviewer 2:

We thank the reviewer for his/her kind comments and were glad to learn that the reviewer felt the manuscript significantly contributed to body of knowledge on COVID-19.

We greatly appreciate reviewer #2 astute observation point out were the language we selected could be more precise. We agree with the reviewer PCR measures RNA copy number and have made modification to the language in the manuscript replacing viral load with RNA copy number.

Respectfully submitted,

Yuan-Po Tu

March 15, 2023

March 28, 2023

Dr. Yuan-Po Tu
The Everett Clinic
Urgent Care
3901 Hoyt
Everett, WA 98201

Re: Spectrum00064-23R1 (COVID-19 antigen results correlate with the quantity of replication-competent SARS-CoV-2 in a cross-sectional study of ambulatory adults during the delta wave.)

Dear Dr. Yuan-Po Tu:

Your manuscript has been accepted, and I am forwarding it to the ASM Journals Department for publication. You will be notified when your proofs are ready to be viewed.

Sincerely,

Oliver Laeyendecker
Editor, Microbiology Spectrum
